# Growth Performance, Meat Quality, Welfare and Behavior Indicators of Broilers Fed Diets Supplemented with *Yarrowia lipolytica* Yeast

Anna Dedousi [1,*], Sotiris I. Patsios [2], Maria-Zoi Kritsa [1], Konstantinos N. Kontogiannopoulos [3], Maria Ioannidou [1], Antonios Zdragas [1] and Evangelia N. Sossidou [1]

1 Veterinary Research Institute, Hellenic Agricultural Organization DIMITRA, GR 57001 Thessaloniki, Greece
2 Laboratory of Natural Resources and Renewable Energies, Chemical Process & Energy Resources Institute (CPERI), Centre for Research and Technology-Hellas (CERTH), Thermi, GR 57001 Thessaloniki, Greece
3 Ecoresources P.C., 3 Kolchidos Str., GR 55131 Thessaloniki, Greece
* Correspondence: dedousi@vri.gr; Tel.: +30-2310365381

**Abstract:** This study investigated the dietary impact of dried *Yarrowia lipolytica* yeast (YLP) on the growth performance, meat quality, welfare and behavior indicators of broilers. It was performed in a commercial poultry farm using 108 13 day-old Ross 308 male broilers. The chicks were randomly and equally divided into three dietary groups CON, YLP3 and YLP5, according to the incorporation rate of YLP in the feed (0%, 3%, and 5%, respectively). A positive effect on foot pad dermatitis (FPD) of YLP-fed broilers was observed without any adverse effects on welfare, behavior, meat quality and the overall growth performance of the broilers. YLP significantly decreased the malondialdehyde (MDA) values in breast and thigh meat. YLP3 birds presented a superior nutrient quality of breast meat, as indicated by the increased concentration of monounsaturated fatty acids (MUFAs) and polyunsaturated fatty acids (PUFAs), decreased levels of saturated fatty acids (SFAs), a better PUFA/SFA ratio and improved health lipid indices. A significant elevation of n-3 PUFAs was observed in the thigh meat of YLP-fed groups, compared to the CON groups. A positive effect on the overall sensory acceptance of thigh meat was detected in the YLP5 group. YLP feeding, at the rate of 3%, seems to be beneficial for improving the meat nutrition quality.

**Keywords:** *Yarrowia lipolytica*; broilers; sustainable protein feed supplement; growth performance; welfare; meat quality

## 1. Introduction

Proteins are essential for a balanced diet, and given the significant rise of the global population in the near future, much larger quantities of food-grade protein will be needed [1]. Foods of animal origin are needed for global food security, since they comprise 25% of the protein intake worldwide [1]. Protein feed stocks are expensive and limited in animals' ratios. Currently, soybean meal, after soy oil extraction, is the main source of vegetable protein for animal feed [2]. In particular, approximately 97% of the globally produced soybean meal is used as animal feed [3]. However, feeding animals with soybean proteins and other grains increases the dependence of animal production on human-edible plants, negatively affecting human food security and sustainability [4].

Therefore, there is an urgent need to use alternative, more sustainable protein feed sources to partly replace the current supply chains and reduce the environmental footprint of animal products [2]. Moreover, such a necessity arises from the fact that traditional sources of protein for animal feed compete with human food for the use of high-quality land for agricultural uses. Thus, alternative animal feed proteins, such as protein-rich food industry by-products, macro-algae, insect meal, and single cell proteins (SCPs), have a

high potential as sustainable protein sources, since they have a low footprint, and can be produced locally on low-quality agricultural land. [5].

YLP is an oleaginous, non-pathogenic yeast, which can be utilized as a SCP in animal feed [6]. Among others, this is due to its capability to produce considerable levels of proteins and lipids from low cost substrates [7]. In addition, these strains can grow in biofuel-produced by-products, such as glycerol, to produce a yeast biomass rich in proteins and/or lipids [8]. The annual production of SCPs reaches 1200 tons of dry mass, that contains approximately between 41% and 45% $w/w$ of protein [9]. Available research evidence shows that the YLP biomass has a high nutritional value. It is a source of high quality proteins (especially essential amino acids), minerals, vitamins and polyunsaturated fatty acids [8,10]. Currently, the American Food and Drug Administration, has examined the safety issues of YLP yeast, and has characterized both the yeast and its various fermentation products as being generally recognized as safe (GRAS) [11]. Moreover, a YLP strain grown on raw glycerol has been considered safe for use as a high value foodstuff (EU 2017/1017) [12]. Finally, the European Food and Safety Authority (EFSA) declared the YLP yeast biomass as a novel food (NF) safe for use, pursuant to Regulation (EU) 2015/2283 [13] on dietary supplements intended for the general population over three years of age [14].

Previous investigations using YLP as a feed supplement for turkeys [15–20], piglets [8,21–23], calves [24], rats [25], fish [26–29], crustaceans [30,31] and mollusks [32,33] have been reported, giving promising results, in terms of their health and performance. More specifically, the incorporation of YLP in the diet of productive animals has been shown to beneficially affect: (1) weight gain, (2) feed conversion ratio, (3) intestinal and ruminal microbiomes, (4) intestinal morphology, (5) antioxidant status and immune response, (6) erythropoietic processes and hematobiochemical profiles, (7) survival rate, (8) digestibility, (9) pathogen elimination (10) the fatty acid composition of fish fillets and (11) the metabolic status of the organism [34]. However, to our knowledge, there are no indications in the available literature regarding the dietary effect of YLP yeast in broilers' growth performance or meat quality. Moreover, no data are available on whether the use of feed with YLP yeast affects the welfare and behavior of productive animals, in general, and broiler chickens, in particular.

This study aims to assess the dietary effect of YLP yeast on broilers' growth performance and meat quality, as well as certain welfare and quality behavior indicators of broilers. Moreover, different inclusion rates of YLP were tested to specify the optimal inclusion rate of YLP in chickens' feed.

## 2. Materials and Methods

### 2.1. Yeast Biomass Production

The yeast biomass, which was used in the feeding trials, was obtained through a series of semi-continuous fermentations of the wild type strain YPL MUCL 28849 (BCCM/MUCL, (Agro) Industrial Fungi & Yeasts Collection, Belgium). All submerged fermentations were conducted using sterilized (121 °C for 20 min) synthetic media supplemented with 46 g/L crude glycerol ($\geq$90–92% $w/w$ purity) that derived as a by-product of the biodiesel production of the Fytoenergeia/NewEnergy S.A. industrial plant, sited in Northern Greece (Paralimnio, Serres). A modified synthetic medium composition was used [35], which comprised (g/L): $(NH_4)_2SO_4$, 3.0; $KH_2PO_4$, 2.0; $Na_2HPO_4 \times 2\ H_2O$, 2.6; $MgSO_4 \times 7\ H_2O$, 1.0; $CaCl_2 \times 2\ H_2O$, 0.2; $FeCl_3$, 0.02; (mg/L): $H_3BO_3$, 0.5; $CuSO_4 \times 5\ H_2O$, 0.06; KI, 0.1; $MnSO_4 \times H_2O$, 0.45; $ZnSO_4 \times 7\ H_2O$, 0.71; and $Na_2MoO_4 \times 2\ H_2O$, 0.23. YPL was transferred from a YPG-agar culture (yeast extract 10 g/L; peptone 20 g/L; glycerol 2% $v/v$) to the first flask preculture for 24-h incubation (30 °C, 150 rpm, 0.1 L). The first preculture content was centrifuged and the pellet was used to inoculate 2 L of a fermentation culture in a bench-scale bioreactor (BioFlo120, Eppendorf, Hamburg, Germany). The bioreactor, which was sterilized at 121 °C for 20 min, was equipped with sensors for optical density, pH, temperature and dissolved oxygen recording. Aeration was kept constant at 0.75 vvm,

the agitation rate was set at 500 rpm, the pH was controlled at 6.0, through the addition of 5 N NaOH and/or 5 N $H_2SO_4$ and the temperature was constant at 30 °C.

Following 20 h of cultivation (middle exponential growth phase), the dry biomass concentration reached approximately 14 g/L, and 1.5 L of the fermentation culture was collected and used as inoculum for the main fermentation in an industrial-scale (140 L) pilot bioreactor (working volume 100 L) that was designed by NRRE Lab (CPERI/CERTH, Thessaloniki, Greece) and operated at the premises of the Fytoenergeia/NewEnergy S.A. industrial plant; a schematic representation of the industrial-scale pilot plant is presented in Figure S1. The reactor was sterilized at 121 °C for at least 20 min, and cooled down at 28 °C, prior to its inoculation. The aeration rate was constant at 1 vvm, the agitation speed was 150 rpm, the pH was controlled at 6.0 through the addition of 30% *w/w* NaOH, and the temperature was set to 28 °C. The main operating parameters, i.e. dissolved oxygen, pH, temperature and bioreactor volume, were monitored through on-line sensors. The bioreactor operated in batch mode for the first 24 h of cultivation; at that point, the dry biomass reached a concentration of approximately 15 g/L. Thereafter, a semi-continuous operation was employed, through the withdrawal of 50 L of culture, and the addition of 50 L of fresh fermentation medium every 24 h. The yeast biomass was recovered from the withdrawn culture through gravity filtration using a fiber-filter bag with a 5 μm pore size (Fluxflo G1PE5-S, Envirogen Group, Alfreton, UK). The wet yeast cake (moisture content approximately 85% *w/w*) was scraped off the filter's surface and subsequently air-dried in an oven at 60 °C, till constant weight. All of the dried yeast biomass collected each day was thoroughly mixed/ homogenized and was used in the feeding trials.

A sample of the homogenized dry yeast biomass was analyzed for its physicochemical properties (nutrients analysis), heavy metals (arsenic, cadmium, lead and mercury), fatty acid and amino acid profiles, and microbiological characterization (Table 1). The physicochemical analysis for the basic nutrients was performed according to the European Commission's (EC) Regulation No 152/2009 [36] for the following parameters: dry matter, crude protein, ash, crude fats and crude fiber; the carbohydrate content and the gross energy were calculated, based on the proximate analysis. The sugar content was measured using an enzymatic method used for the analysis of D-glucose, D-fructose and sucrose in plant and food products, employing a Megazyme K-SUFRG 04/18 assay kit [37]. The amino acid profiles (excluding tryptophan) were also determined, according to the EC's Regulation No 152/2009 [36].

The concentrations of the mono-, poly-unsaturated, and saturated fatty acids were measured through a GC-FID analysis. The extracted fatty acids, via the Soxhlet extraction method [38] (Soxtherm SOX412-MACRO by Gerhardt), were analyzed by GC-FID after transesterification employing a methanol-potassium hydroxide solution. The chromatographic analyses were carried out with a Shimadzu GC-2010 Plus High-End gas chromatography system equipped with an FID detector, employing Supelco SP2560, a 100 m × 0.25 mm × 0.20 μm column, and helium (grade 99.999%) as a carrier gas, at a flow rate of 2 mL/min, according to a previously described analytical protocol [39].

Heavy metals, namely arsenic, cadmium, lead and mercury were determined with ICP-MS. The analysis of heavy metals was carried out via an inductively coupled plasma mass spectrometer (ICP-MS) (Agilent 7850), based on the U.S. Food and Drug Administration [40]. An amount of $HNO_3$ and $H_2O_2$ was added to an aliquot of the sample, followed by a digestion process at 210 °C. Once digestion was completed, the sample was diluted and measured by the ICP-MS method.

The dried yeast biomass and the feed ratios were also microbiologically characterized; the following analyses were made: Enterobacteriaceae (ISO 21528:2004) [41], *E. coli* (ISO 16649:2001) [42], yeasts and molds (ISO 7954:1987) [43], *Salmonella* spp. (ISO 6579:2002) [44] and *Listeria monocytogenes* (ISO 11290:2017) [45].

**Table 1.** Nutrients analysis, heavy metals, fatty acid composition and microbiological characterization of the dried YLP used in the study.

| Items | Dried YLP |
|---|---|
| Moisture and Volatiles (g/100 g) | 3.21 |
| Ash (g/100 g) | 10.96 |
| Fat (g/100 g) | 5.52 |
| Proteins (g/100 g) | 48.77 |
| Crude Fibers % (g/100 g) | 2.20 |
| Carbohydrates (g/100 g) | 29.34 |
| Sugars (g/100 g) | <0.30 * |
| Energy (kcal/100 g) | 362.1 |
| Lysine % | 11.4 |
| Threonine % | 5.6 |
| **Fatty Acids (% of total fats)** | |
| Eicosapentaenoic acid (C20:5 n3) | ND |
| Behenic acid (C22:0) | ND |
| Linoleic acid (C18:2 n6c) | 35.2% |
| Arachidic acid (C20:0) | ND |
| Eicosenic acid (C20:1) | 0.9% |
| α -linolenic acid (C18:3 n3) | 4.5% |
| Margaric acid (C17:0) | 0.5% |
| Heptadecenoic acid (C17:1) | 1.6% |
| Stearic acid (C18:0) | 2.2% |
| Palmitic acid (C16:0) | 9.6% |
| Elaidic acid (C18:1 n9t) | 0.5% |
| Oleic acid (C18:1 n9c) | 42.5% |
| Pentadecanoic acid (C15:0) | 0.5% |
| Palmitoleic acid (C16:1) | 1.8% |
| **Heavy Metals (mg/kg)** | |
| Arsenic | 0.23 |
| Cadmium | 0.004 |
| Lead | <0.04 * |
| Mercury | <0.10 * |
| **Microbiological Characterization (cfu/g)** | |
| Enterobacteriaceae | $4.8 \times 10^6$ |
| *E. coli* | $2.1 \times 10^5$ |
| Yeasts and Molds | $6.8 \times 10^8$ |
| *Salmonella* spp. | ND |
| *Listeria monocytogenes* | ND |

* This value is the detection limit of the assay. ND: not detected.

### 2.2. Animals, Diets and the Experimental Design

A total of 108 13 day-old Ross 308 male broilers, with a starting body weight (BW) of 274.68 ± 1.46 g were used in the present study. The chicks were randomly allocated in 9 consecutive floor pens (12 birds/pen) in a close commercial poultry house in Greece. Each pen was equipped with nipple drinkers and a bell feeder and its floor was covered with rice husk. Throughout the experimental period of 29 days in total, the broiler feed was offered ad libitum, in mash form, and the birds were allowed free access to fresh water. The stoking density in each pen was in agreement with the instructions of EU Directive 2007/43/EC [46]. The temperature, lighting and relative humidity were controlled following the Ross 308 management guidelines (Aviagen 2018) [47].

The broilers were randomly and equally divided into 3 dietary groups CON, YLP3 and YLP5, according to the incorporation rate of YLP in their feed (0%, 3% and 5%, respectively) with 36 chicks/group, 3 replicate-pens/group, 12 chicks/replicate-pen. A three-phase feeding program was used in each dietary treatment, which included a grower diet fed from 13 to 20 days of age, a finisher 1 and a finisher 2 diet, fed from 21 to 32 days of age and from 33 to 41 days of age, respectively. In total, 9 ratios were formulated, 1 per feeding period/dietary treatment (Table 2). In the YLP-diets, the dried yeast in the form of flour

mainly replaced the soybean meal and a small quantity of sunflower oil from the control diet, so as all ratios were isonitrogenous and isocaloric.

**Table 2.** Formulation and nutrient composition of the diets containing YLP, compared with the control diet (CON).

| Items | Grower (13–20 Days) | | | Finisher 1 (21–32 Days) | | | Finisher 2 (33–41 Days) | | |
|---|---|---|---|---|---|---|---|---|---|
| | CON | YLP3 | YLP5 | CON | YLP3 | YLP5 | CON | YLP3 | YLP5 |
| **Ingredients** | | | | | | | | | |
| Wheat | 47.15 | 46.88 | 47.03 | 36.583 | 36.743 | 36.593 | 40.46 | 40.59 | 39.15 |
| Corn | 15.0 | 15.2 | 15.03 | 30.0 | 29.9 | 30.08 | 30.0 | 29.95 | 31.4 |
| Soybean Meal 46.5% | 30.43 | 27.7 | 25.85 | 25.93 | 23.1 | 21.23 | 21.93 | 19.1 | 17.31 |
| Yarrowia Lipolytica | 0 | 3.0 | 5.0 | 0 | 3.0 | 5.0 | 0 | 3.0 | 5.0 |
| Sunflower Oil | 4.4 | 4.2 | 4.07 | 5.0 | 4.77 | 4.61 | 5.0 | 4.75 | 4.53 |
| MCP * | 0.4 | 0.4 | 0.4 | 0.18 | 0.18 | 0.18 | 0.14 | 0.14 | 0.14 |
| Limestone | 1.08 | 1.08 | 1.08 | 0.89 | 0.89 | 0.89 | 0.92 | 0.92 | 0.92 |
| NaCl | 0.27 | 0.27 | 0.27 | 0.27 | 0.27 | 0.27 | 0.27 | 0.27 | 0.27 |
| Dl-Methionine | 0.34 | 0.34 | 0.34 | 0.29 | 0.29 | 0.29 | 0.25 | 0.25 | 0.25 |
| L-Lysine | 0.28 | 0.28 | 0.28 | 0.26 | 0.26 | 0.26 | 0.25 | 0.25 | 0.25 |
| L-Threonine | 0.07 | 0.07 | 0.07 | 0.07 | 0.07 | 0.07 | 0.06 | 0.06 | 0.06 |
| RONOZYME® HiPhos | 0.02 | 0.02 | 0.02 | 0.015 | 0.015 | 0.015 | 0.01 | 0.01 | 0.01 |
| Premix [1] | 0.4 | 0.4 | 0.4 | 0.4 | 0.4 | 0.4 | 0.4 | 0.4 | 0.4 |
| Mycotoxins Binder | 0.1 | 0.1 | 0.1 | 0.01 | 0.01 | 0.01 | 0.2 | 0.2 | 0.2 |
| Enzymes | 0.01 | 0.01 | 0.01 | 0.001 | 0.001 | 0.001 | 0.01 | 0.01 | 0.01 |
| Coccidiostat | 0.05 | 0.05 | 0.05 | 0.001 | 0.001 | 0.001 | 0.05 | 0.05 | 0.05 |
| Lipidol Ultra 0.075% | | | | 0.05 | 0.05 | 0.05 | 0.05 | 0.05 | 0.05 |
| Avatec 150 (150 g/kg) | | | | 0.05 | 0.05 | 0.05 | | | |
| Total | 100 | 100 | 100 | 100 | 100 | 100 | 100 | 100 | 100 |
| **Chemical Analysis** | | | | | | | | | |
| Crude Protein (%) | 20.97 | 20.99 | 21.01 | 18.80 | 18.80 | 18.79 | 17.32 | 17.32 | 17.31 |
| Crude Fiber (%) | 2.91 | 2.77 | 2.68 | 2.81 | 2.67 | 2.58 | 2.70 | 2.57 | 2.48 |
| Fat (%) | 6.1 | 5.90 | 5.77 | 7.05 | 6.82 | 6.67 | 7.05 | 6.80 | 6.62 |
| Ash (%) | 3.0 | 3.22 | 3.37 | 2.67 | 2.89 | 3.04 | 2.67 | 2.89 | 3.04 |
| **Calculated Analysis** | | | | | | | | | |
| ME (kcal/kg) | 2939 | 2941 | 2943 | 3050 | 3051 | 3052 | 3077 | 3077 | 3077 |
| Lysine (%) | 1.28 | 2.19 | 2.79 | 1.13 | 2.04 | 2.64 | 1.02 | 1.93 | 2.53 |
| Methionine (%) | 0.64 | 0.84 | 0.97 | 0.57 | 0.77 | 0.90 | 0.51 | 0.71 | 0.84 |
| Ca (%) | 0.84 | 0.96 | 1.04 | 0.72 | 0.84 | 0.92 | 0.70 | 0.82 | 0.90 |
| P (%) | 0.64 | 0.75 | 0.83 | 0.56 | 0.67 | 0.74 | 0.51 | 0.63 | 0.70 |
| Na (%) | 0.14 | 0.62 | 0.94 | 0.14 | 0.62 | 0.94 | 0.14 | 0.62 | 0.94 |

[1] Premix Finisher contains (per kg of product): vitamin A, 2,500,000 IU; vitamin D3, 1,250,000 IU; vitamin E, 20,000 mg; vitamin K3, 1500 mg; biotin, 35,000 mcg; folic acid, 300 mg; vitamin B1, 1500 mg; vitamin B2, 1500 mg; vitamin B6, 750 mg; vitamin B12, 6000 mcg; niacinamide, 7500 mg; calcium D-pantothenate, 3750 mg; choline chloride, 150,000 mg; carbonate (siderite), 12,500 mg; copper as copper sulphate pentahydrate, 2500 mg; manganic oxide, 27,500 mg; zinc oxide, 20,000 mg; calcium iodate anhydrous, 300 mg; coated granulated sodium selenite, 75 mg; citric acid, 14 mg; orthophosphoric acid, 3.50 mg; butylhydroxytoluene (BHT), 35 mg; butylated hydroxyanisole (BHA), 8.75 mg; calcium carbonate, 55.90%; calcium, 22.21%; phosphorous, 0.01%; * MCP: monocalcium phosphate.

At the end of the trial (day 41), all birds were transported and slaughtered at the local abattoir. Prior to their transportation, 18 chickens (6 birds per group) were randomly selected for meat quality analyses and sensor evaluation testing (SET) and were individually marked (leg bands) for identification. At the abattoir, the carcasses from the selected birds were scalded at 61–65 °C for 60 s, defeathered in a rotary drum picker for 25 s and the whole carcasses (head, feet, blood, without intestines) were air-chilled at 4 °C. Following the chilling process, the selected carcasses were weighed 24 h post-mortem. From each carcass, half of the breast and one thigh were used for meat quality analyses and the other half breast and thigh were used for SET. This way, 12 final breast meat samples/group (6 for meat quality and 6 for SET) and 12 final thigh meat samples/group (6 for meat quality and

6 for SET) were formulated. All meat samples were individually packed in sealable food bags and frozen at $-18\ ^\circ$C temperature until the day of the meat quality analyses and SET.

### 2.3. Production Traits

The broilers' BW was determined at the onset of the experiment (at 13 days old) and at 20, 27 and 41 days old. The body weight gain (BWG) and feed consumption (FC) per bird were measured every week and calculated at weekly intervals, as well as for the whole experimental period (13–41 days of age). Based on the FC and BWG, the feed conversion ratio (FCR) per bird was calculated both at weekly intervals and during the entire experiment. The mortality rate was recorded daily.

### 2.4. Welfare and Behavior Indicators

At the age of 41 days, the chickens from each treatment group were individually evaluated for feather cleanliness, foot pad dermatitis (FPD), hock burn, as well as for quality behavior traits, according to the Welfare Quality (2009) protocol [48]. Initially, observations of the quality behavior characteristics (active, eating, fearful, calm, friendly and pecking behaviors) were undertaken in order to avoid confounding data due to handling stress. For behavioral quality traits, individual visual observations were made lasting 2 min for each pen. In every group, the number of birds exhibiting a particular type of behavior was noted and then divided by the total number of live birds in that group and multiplied by 100. Thus, the data % was calculated. The same methodology was used to estimate the welfare characteristics for each score category/welfare parameter. All observations were made by the same evaluator at 10 a.m.

Then, each bird was gently caught by one person from the research team (same for all assessments) and was examined for the feather cleanliness assessment, by scoring on a 3-point scale: Score 0-completely clean feathers; Score 1-slight feather soiling; Score 2-moderate feather soiling; and Score 3-severe feather soiling. The percentage of chicks presenting each score was then calculated. Both legs of the birds were examined for the presence of FPD (swelling-bubble foot) or hock burn and the scores were estimated according to the following scale: (a) FPD: Score 0-no evidence of FPD; Score 1 and 2-minimal evidence of FPD, Score 3 and 4-evidence of FPD (b) Hock burn: 0-no evidence of hock burn; Score 1 and 2-minimal evidence of hock burn, Score 3 and 4-evidence of hock burn. The percentage of birds with each scoring category was then recorded. Apart from day 41, the birds' quality behavior traits were also estimated at 20 and 27 days of age.

### 2.5. Meat Analysis
### 2.5.1. Materials and Reagents

The following reagents were used in the analyses: sulfuric acid ($H_2SO_4$) 98% for analysis (PanReac AppliChem, Barcelona, Spain, Belgium), sodium hydroxide NaOH (Merck, Darmstadt, Germany), boric acid ($H_3BO_3$) (PanReac AppliChem, Darmstadt, Germany), nitric acid ($HNO_3$) 65% (ChemLab, Zedelgem, Belgium), Tashiro's indicator solution (Honeywell Fluka, Munchen, Germany), hydrochloric acid (HCl) 0.1 N (VWR Chemicals BDH, Rosny-sous-Bois cedex, France), petroleum ether 40–60 ar (ChemLab), boron trifluoride methanol solution (FLuca), n-hexane pesticide grade (ChemLab, Zedelgem, Belgium), tablets Kjeldahl Cu (Gerhardt, Königswinter, Germany), potassium hydroxide (KOH) 85% (Panreac, Barcelona, Spain), methanol (VWR chemicals, Radnor, PA, USA), 2-thiobarbituric acid (Sigma-Aldrich, Darmstadt, Germany), butylated hydroxytoluene (Sigma Aldrich, Darmastadt, Germany) and trichloroacetic acid (Merck, Darmstadt, Germany) [49].

The reference standards used included: a 37 component mixture of fatty acids methyl esters, FAME mix $C_6$–$C_{24}$ (SIGMA 18919-1AMP, certified reference material), a 13 polyunsaturated fatty acids mixture, PUFA No.1 (marine source analytical standard from SIGMA, 47033 with C14:0, C16:0, C16:1 n7, C18:1 n9, C181 n7, C18:2 n6, C20:1 n9, C18:4 n3, C22:1 n11, C22:1 n9, C20:5 n3, C22:5 n3, C22:6 n3), a 14 polyunsaturated fatty acids mixture PUFA No.2 (animal source analytical standard, SIGMA 47015-U with C14:0, C16:0, C16:1 n7, C18:0,

C18:1 n9, C18:1 n7, C18:2 n6, C18:3 n6, C18:3 n3 C20:1 n9, C20:2 n6, C20:3 n6, C20:4 n6, C20:5 n3, C22:4 n6, C22:5 n3 C22:6 n3) and a linoleic acid, conjugated methyl ester standard (CLA), purchased from Sigma Aldrich (SIGMA 05632) (St. Louis, MO, USA) [49].

### 2.5.2. Physicochemical Analysis

The chicken meat samples were analyzed for pH, protein, total nitrogen and lipid oxidation. The pH was measured with an electronic pH-meter (Consort, Belgium). The total nitrogen (TN) of the chicken meat samples was analyzed using the Kjeldahl method, according to the AOAC official method [50]; the raw protein content was estimated as 6.25 times the TN. A duplicate analysis was performed for all parameters under consideration.

### 2.5.3. Lipid Oxidation

Lipid oxidation was determined, based on the formation of malondialdehyde (MDA), using a selective third-order derivative spectrophotometric method [51]. The samples were blended in a small food processor. The subsamples (2 g) were homogenized with 8 ml of aqueous trichloroacetic acid (5% *w/v*) and 5 Ml of butylated hydroxytoluene in hexane (0.8% *w/v*) and the mixture was centrifuged. The top hexane layer was discarded and the bottom aqueous layer was transferred to a volumetric flask (10 mL) and analyzed, according to Ioannidou et al. [49]. Lipid oxidation is expressed as ng of MDA per g of muscle meat.

### 2.5.4. Fatty Acid Composition

Tissue fat was extracted, according to the Soxhlet method, using a Soxtec 2050 (Foss, Tecator, Denmark) automated system. The fatty acid methyl esters were prepared with boron trifluoride in a methanol solution. An appropriate quantity of the extracted fat was saponified by the addition of NaOH in methanol, followed by heating at 100 °C for 15 min. Fatty acid methyl esters were prepared by incubation at 100 °C for 5 min in a boron trifluoride methanol reagent. The produced fatty acid methyl esters were extracted by the addition of 1 mL hexane, followed by the addition of a saturated solution of potassium hydroxide and vigorous agitation [52]. The fatty acid methyl esters were removed and placed in GC vials. The prepared fatty acid methyl esters were analyzed using an HP 5890 (Hewlett-Packard) gas chromatograph equipped with a split/splitless injector (split mode) and a flame ionization detector (FID), according to Ioannidou et al. [49].

Based on the proportions of particular FAs and their groups, the health quality of the meat breast and thigh lipids was estimated by calculating the atherogenic index (AI), thrombogenic index (TI) and the ratio between the hypocholesterolemic (h)/hypercholesterolemic (H) fatty acids (h/H). The following equations were used to calculate these indices:
Atherogenic index [53]:

$$AI = (4 \times C14{:}0 + C16{:}0 + C18{:}0)/(\Sigma MUFA + \Sigma PUFA\text{-}n\text{-}6 + \Sigma PUFA\text{-}n\text{-}3) \tag{1}$$

Thrombogenic index [54]:

$$TI = (C14{:}0 + C16{:}0 + C18{:}0)/(0.5 \times \Sigma MUFA + 0.5 \times \Sigma PUFA\text{-}n\text{-}6 + 3 \times \Sigma PUFA\text{-}n\text{-}3 + \Sigma PUFA\text{-}n\text{-}3/\Sigma PUFA\text{-}n\text{-}6) \tag{2}$$

Ratio between the hypocholesterolemic and hypercholesterolemic fatty acids [55]:

$$h/H = C18{:}1n9c + C18{:}2n6c + C18{:}3n3c + C18{:}3n6c + C20{:}2n6 + C20{:}3n6 + C20{:}4n6 + C22{:}6n3/C14{:}0 + C16{:}0 \tag{3}$$

where:

$\Sigma$ = Summary,
MUFAs = monounsaturated FAs and
PUFAs = polyunsaturated FAs

### 2.6. Sensory Evaluation Testing

2.6.1. Participants

All participants took part voluntarily and signed their informed consent before the SET. All data collected during the SET were irreversibly anonymous, e.g., the individual persons could not be identifiable from the collected data sets. Two groups of consumers were randomly formulated from an initial group of participants to evaluate the broilers' breast and thigh meat, respectively. All participants declared that: (i) their smell and taste were not debilitated (e.g., due to an illness) at the time of the SET; (ii) they consumed broiler meat regularly; and (iii) that they were not allergic to broiler meat or any of the foods used during the SET. The first group comprised 6 persons (4 males and 2 females) aged between 28 and 65 years of age; the second group consisted of 5 persons (3 males and 2 females) aged between 28 and 65 years. All participants were informed about the aims and scope of the SET and were provided with a short explanation of the SET procedure and with instructions on the completion of the evaluation sheet and the interpretation of the 5-point hedonic scale that was used to evaluate the meat samples.

2.6.2. Meat Samples

The meat samples were defrosted overnight in a commercial refrigerator at 4 °C and subsequently cooked unseasoned with the skin removed and any visible external fat trimmed off, in a convection oven at approximately 200 °C. Cooking lasted approximately 45 min until the internal temperature of the meat samples reached 72 °C; the temperature was measured with an electronic hand-held thermometer (model: pH-meter CP-411, company: ELMETRON, country: Poland). Once cooked, the broiler's breasts and thighs were cut into rather uniform meat samples (approximately weight $15 \pm 5$ g), containing no bones, they were randomly placed on a white ceramic food plate, and were immediately served to the participants after reaching a temperature of approximately 60 °C.

2.6.3. Sensory Evaluation Testing

The SET took place the same day for both the breast and thigh meat at the premises of a commercial restaurant in Pieria, Northern Greece. The tasting area was used exclusively for the SET, and each participant was individually and simultaneously served a food plate that contained 3 meat samples (either breast or thigh meat) from each of the three experimental treatments, i.e., CON, YLP3 and YLP5. The placement of the meat samples on the food plate was the same for all participants, who were not aware of the meat samples' identity (i.e., to which experimental treatment they belonged). The participants were allowed to randomly select the order of evaluating the nine meat samples. Following the assessment of each sample, the participants neutralized their taste with a bite of wheat toast containing no salt or sugar, and a sip of natural mineral water. A standard evaluation form was provided to all participants, who manually completed their scores employing a 5-point hedonic scale, from 1: extreme dislike to 5: extreme like. The following characteristics were evaluated for each meat sample: color, flavor, tenderness, juiciness and overall impression.

### 2.7. Statistical Analysis

The statistical analysis of the data was performed using Jeffreys's Amazing Statistics Program JASP (JASP v 0.16.3) software [56]. The significance of the differences of the welfare and behavior indicators among the dietary groups was assessed by a Chi-square test. The normality of the data for the analysis of the broilers' growth performance (BW, BWG, FC and FCR), for the meat quality traits (MDA, meat protein and fat content, fatty acid composition and health lipid indices of the meat) and for the SET were tested employing a Shapiro–Wilk test; the homogeneity of variance was evaluated with Levene's test. A one-way ANOVA was used to compare the average values of the parameters evaluated among the dietary treatments. A post hoc analysis was performed using Tukey's test. When the distribution was not normal, the non-parametric tests Kruskal–Wallis and Mann–Whitney were used to make the comparisons at a significance level of $p \leq 0.05$. A multiple regression

analysis (linear and quadratic) was applied for the evaluation of the relationships between all examined sensory characteristics and the broiler meat's overall acceptance (both for breast and thigh meat). These results are presented as standardized regression coefficients.

## 3. Results

### 3.1. Performance Parameters

The supplementation of YLP in the broilers' diet, did not significantly affect the BWG of the birds ($p > 0.05$), both at weekly intervals and for the whole experimental period (13–41 days of age). The overall growth performance was also not affected by the addition of YLP in their diets, since similar BW results of the chicks at 41 days of age, and FC and FCR, for the whole experimental period ($p > 0.05$), were observed among the groups (Table 3). The highest BW at 20 days of age, was recorded in the broilers fed with YLP at the rate of 3%, followed by the controls and the chickens fed diets with 5% YLP. The observed differences in the BW at 20 days of age were significant only between the YLP3 and YLP5 groups ($p < 0.05$). During the grower phase, the YLP-fed broilers presented a better FCR, compared to the CON chicks, however the observed differences were significant between the CON and YLP3 groups ($p < 0.05$). In the same period, the FC did not differ among the groups ($p > 0.05$). From 21 to 27 days of age, the YLP-fed broilers consumed significantly less feed, compared to the control groups ($p < 0.05$). However, during the grower phase, a similar FCR was recorded among the groups ($p > 0.05$). Moreover, the addition of YLP in the broilers' feed did not affect their BW at 27 days of age ($p > 0.05$). From 28 to 41 days of age, none of the evaluated growth parameters differed among the dietary groups ($p > 0.05$).

**Table 3.** Growth performance of the broilers fed the control and diets containing different levels of YLP. Data are presented as mean $\pm$ SE.

|  | CON | YLP3 | YLP5 |
|---|---|---|---|
| **Body weight (g)** | | | |
| Day 13 | $274.03 \pm 2.74$ | $273.89 \pm 2.52$ | $276.11 \pm 2.38$ |
| Day 20 | $687.08 \pm 12.31$ [ab] | $719.31 \pm 9.40$ [a] | $679.03 \pm 11.49$ [b] |
| Day 27 | $1297.92 \pm 25.50$ | $1320.83 \pm 18.84$ | $1262.36 \pm 20.10$ |
| Day 41 | $2692.92 \pm 54.77$ | $2717.43 \pm 47.26$ | $2627.00 \pm 48.83$ |
| **Body weight gained (g)** | | | |
| 13–20 d | $413.06 \pm 13.62$ | $445.42 \pm 15.22$ | $402.92 \pm 9.17$ |
| 21–27 d | $610.833 \pm 15.34$ | $601.528 \pm 13.66$ | $583.333 \pm 12.09$ |
| 28–41 d | $1395.00 \pm 37.37$ | $1402.29 \pm 36.19$ | $1364.57 \pm 34.78$ |
| Total period (13–41 d) | $806.30 \pm 150.06$ | $817.28 \pm 149.43$ | $784.11 \pm 148.18$ |
| **Feed consumption (g)** | | | |
| 13–20 d | $582.92 \pm 10.49$ | $587.36 \pm 28.32$ | $537.08 \pm 5.29$ |
| 21–27 d | $872.50 \pm 11.47$ [a] | $821.53 \pm 14.87$ [b] | $810.42 \pm 8.32$ [b] |
| 28–41 d | $2005.14 \pm 18.24$ | $2008.03 \pm 13.50$ | $2001.84 \pm 41.61$ |
| Total period (13–41 d) | $1153.52 \pm 217.08$ | $1138.97 \pm 220.11$ | $1116.45 \pm 225.18$ |
| **FCR** | | | |
| 13–20 d | $1.41 \pm 0.03$ [a] | $1.32 \pm 0.02$ [b] | $1.33 \pm 0.02$ [ab] |
| 21–27 d | $1.43 \pm 0.03$ | $1.37 \pm 0.04$ | $1.39 \pm 0.04$ |
| 28–41 d | $1.44 \pm 0.02$ | $1.44 \pm 0.04$ | $1.47 \pm 0.01$ |
| Total period (13–41 d) | $1.43 \pm 0.01$ | $1.37 \pm 0.04$ | $1.40 \pm 0.02$ |

[a,b] Means within a row at a particular age with different superscripts differ significantly ($p < 0.05$).

### 3.2. Welfare and Behavior Indicators

Data regarding the dietary impact of YLP on the broilers' feather cleanliness, FPD and hock burns at the 41st day of their life, are presented in Table 4. The percentage of broilers with completely clean feathers (Score 0) or slight feather soiling (Score 1) did not significantly differ among the dietary groups ($p > 0.05$). However, a significantly higher percentage of birds with moderate feather soiling (Score 2) was recorded in the YLP5 group, compared to that observed in the CON and YLP3 groups ($p < 0.05$). Interestingly, a significantly higher percentage of broilers with no evidence of FPD (Score 0) was observed in the

YLP-fed broilers, compared to the controls ($p < 0.05$). The evaluation of the 41 day-old chicks for evidence of hock burns revealed no significant differences among the experimental groups ($p > 0.05$).

**Table 4.** Percentage of broilers scoring for the welfare parameters (feather cleanliness, FPD, hock burn) at the age of 41 days, in regard to the three dietary treatments (CON, YLP3, YLP5).

| | | Day 41 | | |
|---|---|---|---|---|
| | Score [1] | CON | YLP3 | YLP5 |
| | | **Feather Cleanliness** | | |
| | 0 | 52.78 | 51.43 | 31.42 |
| | 1 | 44.44 | 48.57 | 34.29 |
| | 2 | 2.78 [a] | 0.00 [a] | 34.29 [b] |
| | 3 | | | |
| | | **Foot Pad Dermatitis** | | |
| | 0 | 63.89 [a] | 91.43 [b] | 82.86 [a] |
| | 1 | 5.56 | 5.71 | 11.43 |
| | 2 | 19.44 | 2.86 | 5.71 |
| | 3 | 11.11 | 0.00 | 0.00 |
| | 4 | | | |
| | | **Hock Burn** | | |
| | 0 | 61.11 | 62.86 | 45.71 |
| | 1 | 27.78 | 25.71 | 25.71 |
| | 2 | 5.56 | 8.57 | 14.29 |
| | 3 | 2.78 | 0.00 | 5.72 |
| | 4 | 2.77 | 2.86 | 8.57 |

[a,b] Means within a row at each score category with different superscripts differ significantly ($p < 0.05$). [1] Feather cleanliness: score 0: indicates completely clean feathers, score 1: indicates slight feather soiling, score 2: indicates moderate feather soiling and Score s: indicates severe feather soiling; foot pad dermatitis: score 0: indicates no evidence of FPD, score 1 and 2: indicate minimal evidence of FPD, score 3 and 4: indicate evidence of FPD; hock burn: score 0: indicates no evidence of hock burn, score 1 and 2: indicate minimal evidence of hock burn, score 3 and 4: indicate evidence of hock burn.

Table 5 displays the data of the quality behavior parameters that were evaluated on the 20, 27 and 41 day-old broilers from all three dietary groups. The incorporation of YLP yeast in the broilers' diet significantly affected the feeding behavior of the 20 day-old chicks ($p < 0.05$). In particular, a higher percentage of 20 day-old chicks from the CON group were recorded feeding, in comparison to that observed in the YLP groups, with significant differences noticed between the CON and YLP3 groups ($p < 0.05$). At 27 and 41 days of age, the chicks of all groups expressed similar quality behavioral traits ($p > 0.05$).

**Table 5.** Percentage of broilers observed in the three dietary treatments (CON, YLP3, YLP5), scoring for quality behavior characteristics (active, feeding, fearful, calm, friendly and pecking behaviors) at the age of 20, 27 and 41 day.

| Quality Behavior Traits | Day 20 | | | Day 27 | | | Day 41 | | |
|---|---|---|---|---|---|---|---|---|---|
| | CON | YLP3 | YLP5 | CON | YLP3 | YLP5 | CON | YLP3 | YLP5 |
| Active | 36.11 | 41.67 | 27.78 | 36.11 | 19.44 | 27.78 | 5.55 | 17.14 | 8.57 |
| Feeding | 22.22 [a] | 2.78 [b] | 5.55 [a] | 2.78 | 2.78 | 2.78 | 2.78 | 2.86 | 0 |
| Fearful | 0 | 0 | 0 | 0 | 0 | 0 | 0 | 0 | 0 |
| Calm | 27.78 | 47.22 | 50.00 | 58.33 | 72.22 | 63.89 | 91.67 | 80.00 | 85.71 |
| Friendly | 13.89 | 8.33 | 13.89 | 2.78 | 5.56 | 5.55 | 0 | 0 | 0 |
| Pecking | 0 | 0 | 2.78 | 0 | 0 | 0 | 0 | 0 | 5.72 |

[a,b] Means within a row at a particular age for each type of behavior with different superscripts differ significantly ($p < 0.05$).

### 3.3. Meat Analysis

As demonstrated in Table 6, the breast and thigh meat chemical composition did not differ among the dietary groups, regarding fat and protein ($p > 0.05$). Moreover, a similar breast and thigh meat pH was recorded among all dietary groups ($p > 0.05$) (Table 6). Moreover, lipid oxidation analysis of the breast and thigh meat (Table 7) revealed significant differences among the dietary treatments ($p < 0.05$). Thigh meat MDA was found significantly higher ($p < 0.05$) in the CON group, compared to the YLP groups. Breast meat MDA was also higher in the CON group, in comparison to the YLP groups, however significant differences ($p < 0.05$) were found only between the CON and YLP3 groups.

**Table 6.** Protein and fat content in the breast and thigh meat of the broilers of the three dietary groups. Data are presented as mean $\pm$ SE.

| Breast | CON | YLP3 | YLP5 |
|---|---|---|---|
| Protein % | 23.18 ± 0.34 | 22.58 ± 0.62 | 23.23 ± 0.47 |
| Fat % | 0.46 ± 0.20 | 0.43 ± 0.11 | 0.33 ± 0.06 |
| pH | 5.68 ± 0.03 | 5.66 ± 0.09 | 5.78 ± 0.05 |
| **Thigh** | | | |
| Protein % | 20.11 ± 0.30 | 20.31 ± 0.22 | 19.85 ± 0.26 |
| Fat % | 0.58 ± 0.09 | 0.78 ± 0.10 | 0.51 ± 0.10 |
| pH | 6.00 ± 0.05 | 6.03 ± 0.06 | 6.04 ± 0.04 |

No significant differences were detected among groups ($p > 0.05$).

**Table 7.** Effect of the CON and YLP diets on the broiler chickens' meat oxidative stability. Data are presented as mean $\pm$ SE.

| MDA ppb (ng/gr) | CON | YLP3 | YLP5 |
|---|---|---|---|
| **Thigh meat lipid oxidation** | | | |
| Day 1 of storage | 87.63 ± 9.37 [a] | 57.36 ± 4.72 [b] | 52.71 ± 8.33 [b] |
| **breast meat lipid oxidation** | | | |
| Day 1 of storage | 46.33 ± 8.27 [a] | 24.81 ± 3.97 [b] | 30.08 ± 5.68 [ab] |

[a,b] Means within a row with different superscripts differ significantly ($p < 0.05$). MDA: malondialdehyde.

The fatty acid (FA) composition analysis of the breast meat revealed that MUFAs, PUFAs, n-6 PUFAs and SFAs, as well as the PUFA/SFA and n-6 PUFA/n-3 PUFA ratios were significantly affected ($p < 0.05$) by the dietary treatments (Table 8). In particular, the YLP3 group presented significantly higher ($p < 0.05$) MUFAs and PUFAs and significantly lower ($p < 0.05$) SFAs, compared to the CON and YLP5 groups, respectively. Similarly, the breast meat of the YLP3 broilers had significantly higher n-6 PUFAs ($p < 0.05$) than that found to the breast meat of the other two groups. Moreover, a numerical increase of n-3 PUFAs was recorded in the YLP groups, compared to CON but the observed differences were not significant ($p > 0.05$). Moreover, the breast meat of the YLP3 group had a significantly higher n-6 PUFA/n-3 PUFA ratio than that recorded in the other two groups ($p < 0.05$). The significant differences of the MUFA, PUFA and SFA levels observed in the breast meat of all investigated groups, resulted in relevant differences of the PUFA/SFA ratio. More specifically, the CON and YLP5 groups had a significantly lower PUFA/SFA ratio than the YLP3 group ($p < 0.05$). The incorporation of YLP in the diet of broilers significantly affected the health lipid indices of the breast meat ($p < 0.05$). As shown in Table 8 the lowest atherogenic and thrombogenic indices were recorded in the YLP3 group and the highest ones in the CON and YLP5 groups ($p < 0.05$). In contrast, the highest h/H ratio was observed in the breast meat of YLP3 group ($p < 0.05$).

**Table 8.** The effect of the YLP supplementation on the broiler breast meat FA composition and health lipid indices. Data are presented as mean ± SE.

| Item | CON | YLP3 | YLP5 |
|---|---|---|---|
| MUFA % | 36.78 ± 0.65 [b] | 41.97 ± 1.81 [a] | 35.46 ± 0.76 [b] |
| PUFA % | 2.84 ± 1.18 [b] | 16.66 ± 4.71 [a] | 4.39 ± 0.92 [b] |
| SFA % | 60.38 ± 1.17 [a] | 41.37 ± 6.03 [b] | 60.15 ± 1.23 [a] |
| PUFA/SFA | 0.05 ± 0.02 [b] | 0.52 ± 0.18 [a] | 0.07 ± 0.02 [b] |
| PUFA n6 | 2.72 ± 1.10 [b] | 16.10 ± 4.55 [a] | 4.04 ± 0.88 [b] |
| PUFA n3 | 0.12 ± 0.09 | 0.35 ± 0.09 | 0.29 ± 0.07 |
| PUFA n6/ PUFA n3 | 12.92 ± 1.08 [b] | 44.13 ± 8.86 [a] | 12.33 ± 2.18 [b] |
| AI | 1.62 ± 0.08 [a] | 0.85 ± 0.23 [b] | 1.64 ± 0.10 [a] |
| TI | 2.90 ± 0.16 [a] | 1.55 ± 0.42 [b] | 2.80 ± 0.14 [a] |
| h/H | 0.80 ± 0.06 [b] | 2.07 ± 0.44 [a] | 0.85 ± 0.06 [b] |
| **Fatty acids** | | | |
| Myristic acid (C14:0) | 1.930 ± 0.234 [a] | 1.017 ± 0.179 [b] | 2.415 ± 0.497 [a] |
| Myristelic acid (C14:1) | 0.168 ± 0.059 | 0.092 ± 0.011 | 0.112 ± 0.025 |
| Pentadecanoic acid (C15:0) | 0.410 ± 0.054 [a] | 0.227 ± 0.029 [b] | 0.477 ± 0.036 [a] |
| Pentadecenoic acid (C15:1) | ND | ND | ND |
| Palmitic acid (C16:0) | 40.922 ± 1.420 [a] | 28.717 ± 3.921 [b] | 39.470 ± 1.384 [a] |
| Palmitoleic acid (C16:1) | 3.024 ± 0.333 [a] | 3.088 ± 0.221 [a] | 1.878 ± 0.081 [b] |
| Margaric acid (C17:0) | 0.512 ± 0.071 | 0.372 ± 0.152 | 0.568 ± 0.077 |
| Heptadecenoic acid (C17:1) | ND | ND | ND |
| Stearic acid (C18:0) | 15.090 ± 0.841 [ab] | 10.358 ± 1.889 [b] | 15.520 ± 0.746 [a] |
| Elaidic acid (C18:1n9t) | 0.424 ± 0.027 [b] | 0.413 ± 0.046 [b] | 0.713 ± 0.084 [a] |
| Oleic (C18:1n9c) | 31.244 ± 0.389 [b] | 36.413 ± 1.650 [a] | 30.752 ± 0.666 [b] |
| Vaccenic acid (C18:1n7) | 1.586 ± 0.104 | 1.562 ± 0.092 | 1.428 ± 0.149 |
| Linolelaidic (C18:2n6t) | ND | ND | ND |
| Linoleic acid (C18:2n6c) | 2.578 ± 1.037 [b] | 15.507 ± 4.407 [a] | 3.658 ± 0.841 [b] |
| γ-Linolenic acid (C18:3n6) | 0.140 ± 0.069 | 0.238 ± 0.065 | 0.377 ± 0.072 |
| α-Linolenic acid (C18:3n3) | 0.090 ± 0.064 | 0.328 ± 0.101 | 0.250 ± 0.070 |
| Conjugated linoleic acid CLA | ND | 0.035 ± 0.087 | 0.067 ± 0.163 |
| Stearidonic acid (C18:4n3) | ND | ND | ND |
| Arachidic acid (C20:0) | 0.494 ± 0.099 [a] | 0.252 ± 0.034 [b] | 0.502 ± 0.066 [a] |
| Gondoic acid (C20:1n9) | 0.338 ± 0.150 | 0.358 ± 0.030 | 0.305 ± 0.098 |
| Eicosadienoic acid (C20:2) | ND | 0.135 ± 0.053 | ND |
| Eicosadienoic acid (C21:0) | 0.120 ± 0.062 [ab] | 0.045 ± 0.021 [b] | 0.232 ± 0.045 [a] |
| Dihomo-γ-linolenic acid (C20:3n6) | ND | 0.090 ± 0.041 | ND |
| Arachidonic acid (C20:4n6) | ND | 0.223 ± 0.081 | ND |
| Eicosatrienoic acid (C20:3n3) | ND | ND | ND |
| Behenic acid (C22:0) | 0.704 ± 0.169 [ab] | 0.302 ± 0.035 [b] | 0.643 ± 0.031 [a] |
| Eicosapentaenoic acid (EPA)(C20:5n3) | ND | ND | ND |
| Docosenoic acid (C22:1n11) | ND | ND | ND |
| Erucic acid (C22:1n9) | ND | ND | 0.290 ± 0.096 |
| Docosadienoic acid (C22:2) | ND | ND | ND |
| Tricosanoic acid (C23:0) | ND | ND | ND |
| Docosatetraenoic acid (C22:4n6) | ND | 0.043 ± 0.028 | ND |
| Lignoceric acid (C24:0) | 0.196 ± 0.090 [ab] | 0.082 ± 0.031 [b] | 0.323 ± 0.082 [a] |
| Docosapentaenoic acid (C22:5n3) | ND | ND | ND |
| Nervonic acid (C24:1) | ND | ND | ND |
| Docosahexaenoic acid (DHA) (C22:6n3) | ND | ND | ND |

[a,b] Means within a row with different superscripts differ significantly ($p < 0.05$), ND: Not Detected,. AI: Atherogenic Index, TI: Thrombogenic Index, h/H: hypocholesterolemic (h)/hypercholesterolemic (H) fatty acids.

The supplementation of YLP in the broilers' diet also affected the individual breast meat FAs. The differences observed among the dietary treatments are presented in detail in Table 8. It seems that most of the individual breast meat FAs, except myristoleic acid (C14:1), margaric acid (C17:0), vaccenic acid (C18:1n7), γ -linolenic acid (C18:3n6), α-linolenic acid (C18:3n3) and gondoic acid (C20:1n9), differed significantly ($p < 0.05$). In general, the most

abundant fatty acid among SFAs recorded in the breast meat of all investigated groups, was the palmitic acid (C16:0), followed by stearic acid (C18:0). The breast meat from the YLP3 group had a significantly lower concentration of palmitic acid, compared to the CON and YLP5 groups ($p < 0.05$). Moreover, the breast meat from the CON and YLP5 groups had higher levels of stearic acid, compared to the YLP3 group however, significant differences were observed only between the YLP groups ($p < 0.05$). A data analysis revealed that oleic acid (C18:1n9c) was the most abundant of the MUFAs detected in the breast meat of all dietary treatments and its concentration was found significantly higher in the YLP3 group ($p < 0.05$), compared to the other two groups. According to the results, linoleic acid (C18:2n6c) was the most abundant of the PUFAs recorded in the breast meat of all groups. The highest concentration of linoleic acid was found in the breast meat of the YLP3 group and differed significantly ($p < 0.05$) from that recorded in the CON and YLP5 groups.

FA composition analysis of the thigh meat demonstrated that the incorporation of YLP in broilers diet did not affect the concentration of MUFA, SFA, PUFA and PUFA n6 FAs as well as the PUFA/SFA and PUFA n6/PUFA n3 ratios ($p > 0.05$) (Table 9). However, significant higher levels of PUFA n3 were detected in the thigh meat of YLP groups compared to CON group ($p < 0.05$). Consequently, few significant differences of individual FAs in the thigh meat of birds were observed among groups. In particular, significantly higher concentration of α-Linolenic acid was detected in the thigh meat of YLP groups compared to CON ($p < 0.05$). CLA and γ-Linolenic acid were detected only in the thigh meat of YLP groups and their concentration was found significantly lower in YLP3 compared to YLP5 group ($p < 0.05$). From the MUFAs detected in the thigh meat of birds, palmitoleic acid (C16:1) was significantly lower in YLP5 compared to CON group ($p < 0.05$). Finally, the health lipid indices of thigh meat did not differ significantly among dietary treatments ($p > 0.05$).

**Table 9.** The effect of YLP supplementation on broiler thigh meat FA composition and health lipid indices. Data are presented as mean ± SE.

| Item | CON | YLP3 | YLP5 |
|---|---|---|---|
| MUFA % | 38.39 ± 1.38 | 36.93 ± 4.56 | 35.54 ± 3.08 |
| PUFA % | 2.71 ± 0.72 | 12.11 ± 4.49 | 13.77 ± 5.37 |
| SFA % | 58.90 ± 2.03 | 50.97 ± 8.63 | 50.69 ± 7.83 |
| PUFA/SFA | 0.05 ± 0.01 | 0.37 ± 0.16 | 0.42 ± 0.21 |
| PUFA n6 | 2.65 ± 0.72 | 11.45 ± 4.35 | 13.03 ± 5.10 |
| PUFA n3 | 0.06 ± 0.04 [a] | 0.45 ± 0.13 [b] | 0.49 ± 0.11 [b] |
| PUFA n6/ PUFA n3 | 25.58 ± 18.23 | 28.49 ± 7.30 | 21.98 ± 5.39 |
| AI | 1.56 ± 0.14 | 1.65 ± 0.61 | 1.46 ± 0.49 |
| TI | 2.80 ± 0.24 | 2.58 ± 0.84 | 2.48 ± 0.81 |
| h/H | 0.83 ± 0.08 | 1.49 ± 0.48 | 1.61 ± 0.53 |
| **Fatty acids** | | | |
| Myristic acid (C14:0) | 1.998 ± 0.286 | 2.265 ± 0.647 | 1.727 ± 0.393 |
| Myristelic acid (C14:1) | 0.114 ± 0.013 | 0.210 ± 0.083 | 0.112 ± 0.016 |
| Pentadecanoic acid (C15:0) | 0.432 ± 0.120 | 0.303 ± 0.061 | 0.410 ± 0.065 |
| Pentadecenoic C15:1) | ND | ND | ND |
| Palmitic acid (C16:0) | 41.054 ± 1.738 | 36.408 ± 6.136 | 34.575 ± 5.296 |
| Palmitoleic acid (C16:1) | 3.368 ± 0.236 [a] | 3.463 ± 0.606 [ab] | 2.562 ± 0.118 [b] |
| Margaric acid (C17:0) | 0.312 ± 0.082 | 0.323 ± 0.085 | 0.417 ± 0.080 |
| Heptadecenoic acid (C17:1) | ND | ND | ND |
| Stearic acid (C18:0) | 13.972 ± 0.750 | 10.928 ± 2.162 | 12.372 ± 1.859 |
| Elaidic acid (C18:1n9t) | 0.582 ± 0.167 | 0.385 ± 0.093 | 0.412 ± 0.046 |
| Oleic (C18:1n9c) | 32.530 ± 1.217 | 30.955 ± 3.870 | 30.623 ± 3.002 |
| Vaccenic acid (C18:1n7) | 1.450 ± 0.089 | 1.373 ± 0.157 | 1.295 ± 0.097 |
| Linolelaidic (C18:2n6t) | ND | ND | ND |

**Table 9.** *Cont.*

| Item | CON | YLP3 | YLP5 |
|---|---|---|---|
| Linoleic acid (C18:2n6c) | 2.574 ± 0.681 | 11.113 ± 4.284 | 12.402 ± 5.059 |
| γ-Linolenic (C18:3n6) | ND | 0.188 ± 0.057 [a] | 0.492 ± 0.101 [b] |
| α-Linolenic (C18:3n3) | 0.060 ± 0.040 [a] | 0.365 ± 0.103 [b] | 0.487 ± 0.105 [b] |
| Conjugated linoleic acid CLA | ND | 0.163 ± 0.077 [a] | 0.203 ± 0.142 [b] |
| Stearidonic acid (C18:4n3) | ND | ND | ND |
| Arachidic acid (C20:0) | 0.468 ± 0.091 | 0.287 ± 0.049 | 0.385 ± 0.070 |
| Gondoic acid (C20:1n9) | 0.342 ± 0.172 | 0.490 ± 0.111 | 0.422 ± 0.122 |
| Eicosadienoic acid (C20:2) | ND | 0.047 ± 0.031 | 0.047 ± 0.031 |
| Eicosadienoic acid (C21:0) | ND | ND | 0.112 ± 0.051 |
| Dihomo-γ-linolenic acid (C20:3n6) | ND | ND | ND |
| Arachidonic acid (C20:4n6) | ND | 0.120 ± 0.054 | 0.105 ± 0.067 |
| Eicosatrienoic acid (C20:3n3) | ND | ND | ND |
| Behenic acid (C22:0) | 0.430 ± 0.185 | 0.312 ± 0.084 | 0.500 ± 0.143 |
| Eicosapentaenoic acid (EPA)(C20:5n3) | ND | ND | ND |
| Docosenoic acid (C22:1n11) | ND | ND | ND |
| Erucic acid (C22:1n9) | ND | ND | ND |
| Docosadienoic acid (C22:2) | ND | ND | ND |
| Tricosanoic acid (C23:0) | ND | ND | ND |
| Docosatetraenoic acid (C22:4n6) | ND | ND | ND |
| Lignoceric acid(C24:0) | 0.238 ± 0.129 | 0.115 ± 0.047 | 0.197 ± 0.083 |
| Docosapentaenoic acid (C22:5n3) | ND | ND | ND |
| Nervonic acid (C24:1) | ND | ND | ND |
| Docosahexaenoic acid (DHA) (C22:6n3) | ND | ND | ND |

[a,b] Means within a row with different superscripts differ significantly ($p < 0.05$), ND: Not Detected, AI: Atherogenic Index, TI: Thrombogenic Index, h/H: hypocholesterolemic (h)/hypercholesterolemic (H) fatty acids.

### 3.4. Sensory Evaluation Test

3.4.1. Breast Meat

The breast meat of broilers fed with different proportions of dried yeast biomass added in their feeding ratios showed some minor differences ($p > 0.05$), as scored by the taste panel, on a scale of 1 to 5 (Figure 1 and Table 10). The highest overall acceptance for breast meat, i.e., with a value of 3.67 ± 0.91, was recorded for broilers fed with 3% of dried yeast biomass, slightly lower than the corresponding acceptance for thigh meat, which scored a value of 4.00 ± 0.76, in the case of broilers fed with 5% of dried yeast biomass.

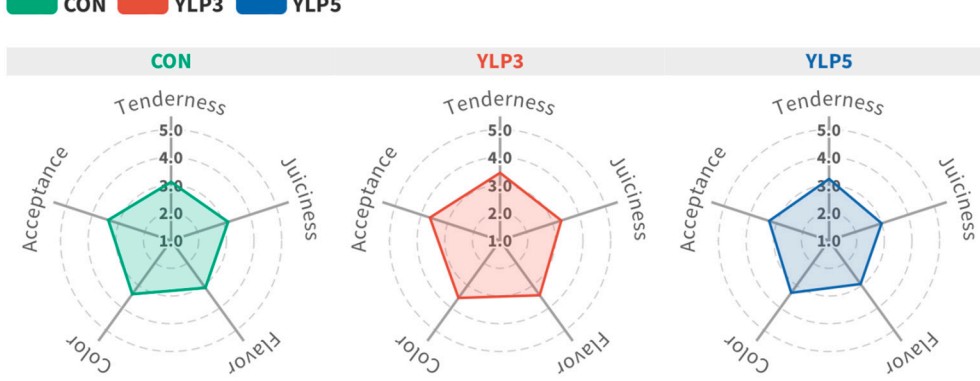

**Figure 1.** Sensory characteristics of broiler breast meat. Characteristics were judged by a panel on a scale from 1 to 5.

A data analysis showed no statistically significant differences ($p > 0.05$), for any sensory characteristics, among the different feeding scenarios for the breast meat. Regarding flavor, the YLP3 broilers scored the highest value (3.44 ± 0.86), meaning that their meat had a slightly better flavor, compared to the other two treatments. Similarly, the breast fillets of

the YLP3 broilers were scored to be more tender ($3.44 \pm 1.10$), juicier ($3.33 \pm 0.97$), and with a better color ($3.56 \pm 0.70$) than the other two treatments; however, the differences cannot be considered statistically significant under the SET conditions ($p > 0.05$). Consequently, regarding breast meat, the YLP3 treatment scored the highest in the overall acceptance and had a higher score in every individual characteristic.

**Table 10.** Data for the statistical means and variability for the broiler meat evaluation.

| | Type of Meat | | | | |
|---|---|---|---|---|---|
| | Tenderness | Juiciness | Flavor | Color | Overall Acceptance |
| | **Breast meat** | | | | |
| CON | $3.11 \pm 0.18$ | $3.17 \pm 0.20$ | $3.11 \pm 0.20$ | $3.39 \pm 0.18$ | $3.39 \pm 0.18$ |
| YLP3 | $3.44 \pm 1.26$ | $3.33 \pm 0.23$ | $3.44 \pm 0.20$ | $3.56 \pm 0.17$ | $3.67 \pm 0.21$ |
| YLP5 | $3.22 \pm 0.27$ | $3.00 \pm 0.23$ | $2.94 \pm 0.15$ | $3.33 \pm 0.18$ | $3.28 \pm 0.23$ |
| | **Thigh meat** | | | | |
| CON | $3.40 \pm 0.29$ | $3.27 \pm 0.25$ | $3.27 \pm 0.21$ | $3.20 \pm 0.28$ | $3.33 \pm 0.21$ [b] |
| YLP3 | $3.87 \pm 0.26$ | $3.67 \pm 0.23$ | $3.40 \pm 0.21$ | $3.73 \pm 0.27$ | $3.53 \pm 0.26$ [ab] |
| YLP5 | $4.13 \pm 0.22$ | $3.93 \pm 0.27$ | $3.53 \pm 0.22$ | $3.87 \pm 0.24$ | $4.00 \pm 0.20$ [a] |

[a,b] Means within a column with different superscripts differ significantly ($p < 0.05$). All values are expressed as mean $\pm$ S.E.

### 3.4.2. Thigh Meat

Evaluating the broilers' thigh meat, the taste panel found statistically significant differences between the three treatments ($p < 0.05$) (feeding scenarios) (as shown in Figure 2 and Table 10). The broilers fed with 5% of dried yeast biomass had the highest overall acceptance, reaching the value of $4.00 \pm 0.76$, while the broilers fed with a CON diet scored the lowest overall acceptance ($3.33 \pm 0.82$). The differences between the other sensory characteristics existed, but they were not statistically significant ($p > 0.05$). Regarding tenderness, the YLP5 broilers scored higher ($4.13 \pm 0.83$), meaning their meat was more tender than those from the other two treatments. Similarly, the thigh fillets of the broilers fed with 5% dried yeast biomass were found to be juicier and with a better color, compared to the other two groups. Summarizing, the YLP5 treatment scored the highest values for all sensory characteristics (only overall acceptance was statistically significant), followed by the YLP3 feeding scenario, while the CON group scored the lowest values. These observations show that the addition of 5% of dried yeast biomass to the feeding ratios of broilers has a positive effect on the sensory characteristics and overall acceptance of the thigh meat. Comparing the breast to thigh meat, the latter scored higher in most cases regardless of the sensory characteristics and feeding treatment.

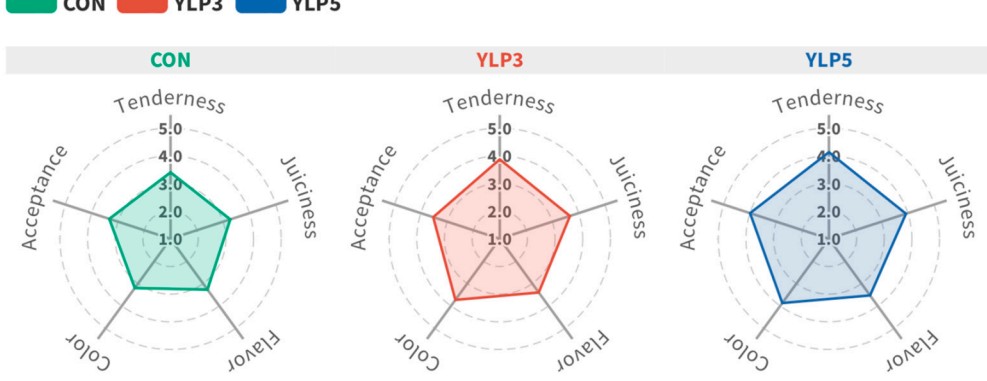

**Figure 2.** Sensory characteristics of broiler leg meat. Characteristics were judged by a panel on a scale from 1 to 5.

### 3.4.3. Regression Analysis

Multiple regression analyses (linear and quadratic) were performed to assess the correlations between the overall acceptance of the broiler breast and thigh meat assortment and its other analyzed sensory characteristics. As shown in Table 11, the tenderness and flavor had the greatest influence on the overall acceptance of broiler breast meat ($p < 0.001$ for linear regression), according to the taste panel. Furthermore, tenderness and color had a considerable effect on the overall acceptance of thigh meat, while juiciness was the least significant factor. In the case of broiler thigh meat, the highest effect upon the overall acceptance of the meat was attributed to meat tenderness ($p < 0.05$ for linear regression) followed by its color ($p = 0.05$ for linear regression). The other sensory characteristics had a lesser influence on the overall acceptance assessment of the meat; however, the flavor sensation had a greater impact than the juiciness. The results from the quadratic regression analysis are summarized in Table S1; however, it did not provide a statistically significant regression between the analyzed sensory characteristics and the overall acceptance.

**Table 11.** Results of the linear regression for the relationships with the broiler overall acceptance (for all types of meat).

| | Breast Meat | | Thigh Meat | |
|---|---|---|---|---|
| | $R^2 = 0.792$ | | $R^2 = 0.629$ | |
| | $b$ [1] | $p$ [2] | $b$ [1] | $p$ [2] |
| Tenderness | 0.489 | <0.001 | 0.44 | 0.015 |
| Juiciness | 0.093 | 0.297 | 0.192 | 0.311 |
| Flavor | 0.342 | <0.001 | 0.151 | 0.227 |
| Color | 0.156 | 0.078 | 0.23 | 0.05 |

[1] Standardized partial coefficient of the linear regression. [2] $p$-value based on the linear regression analysis.

## 4. Discussion

This is the first report that investigates the potential use of YLP yeast in the diet of broilers. Relative previous studies in birds have been carried out only in turkey hens, aged 1–16 weeks old [15–20]. According to our results, the incorporation of YLP did not affect the overall growth performance of chicks, as indicated by the final BW at 41 days-old, and the BWG, FC and FCR during the whole experimental period (13–41 days of age). Only some differences in the BW, FC and FCR of broilers were recorded at weekly intervals among groups. The higher BW of the YLP3 chicks at 20 days of age, compared to the other two groups, could be attributed to the numerical increase of the FC and a net numerical increase of the BWG of birds in this group during the growing period. Moreover, the reduced FC recorded in the YLP groups, compared to the CON group, between 21 and 27 days of age, could be due to an adaption of the birds to the finisher 1 diet. Finally, the observed differences in the feed efficiency between the CON and YLP3 groups during the growing period is the net result of the numerical differences in the FC and BWG seen between the groups during this time period. Similar to our findings, previous reports demonstrated that the supplementation of 3% YLP in turkeys' diet had no impact in the final BW, BWG, FC [17,19] and FCR [17]. However, Czech et al. [19] noted an amelioration of the FCR in YLP3-fed birds, compared to the controls. Moreover, Merska et al. [17] found that increasing the incorporation rate of YLP to turkeys' feed from 3% to 6%, resulted in a lower BW of birds, compared to the CON group and those fed with 3% YLP diets. The higher percentage of CON chicks that were recorded feeding at 20days of age, in comparison to that observed in the YLP groups is probably not related to the YLP supplementation. This is consistent with the non-significant difference in food consumption during the growing phase of birds of all groups. Thus, given the instant character of the quality behavior traits recording procedure, this finding could be considered random.

The incorporation of YLP in broilers' diet at both studied levels (3% and 5%), had a positive effect on FPD, as evaluated at 41 days of age. This finding is considered of

critical importance for chickens' health and welfare, but also for the farmers' financial income. FPD is a multifactorial problem with litter quality, nutrition and gut health as some of the factors implicated in its incidence [57–59]. However, litter condition is considered the most important risk factor for the development of FPD [60]. The litter moisture and ammonia concentration from accumulated fecal material can burn and weaken the dermis of the footpad [61], with an increased severity of FPD resulting from the prolonged exposure of feet to wet litter. Moisture causes the outer layer of the dermis to soften, posing a risk of microbial contamination, leading to necrosis [62]. FPD is a very common and well recognized problem in the broiler industry [58] that negatively affects birds' productivity and welfare [57] and it has been associated with a reduced mobility, lameness and consequently with behavioral restrictions of birds [59,63]. FPD has also been shown to be highly associated with systemic bacterial infections, as pathogens can enter chickens' bodies through the damaged epithelium in the foot pads [64]. Furthermore, financial losses due to FPD are basically attributed to the abattoir condemnation of carcasses with contact dermatitis lesions [58]. Chicken legs is a highly profitable industrial by-product, and poor footpad conditions due to FPD degrade the product quality, leading to rejections and loss of income [59].

The exact mechanism involved in the positive nutritional effect of YLP on FPD observed in this study is presently unknown and requires further investigation as there are no similar studies in the available literature. Two possible explanations could be given for the interpretation of this result. According to the first one, it is possible that YLP ameliorated the gut health of YLP birds, resulting in a reduction of their fecal moisture and consequently reduced litter moisture, which is one of the major causative factors of FPD. The beneficial effect of YLP in intestinal function and gut microbiota has been previously demonstrated in turkey hens [19,20]. The addition of 3% YLP in turkeys' diet increased the intestinal villus length, as well as the ratio of the villus length to the crypt depth and the intestinal muscular layer thickness, thus improving the birds' intestinal health [19]. In another study it was proved that feeding turkey hens with 3% YLP favorably influenced the intestinal microbiota, since it reduced the number of fungi and coliforms, including *Escherichia coli* [20]. In similar studies performed on growing piglets, the incorporation of YLP in the piglets' diet decreased the total number of coliform bacteria and *Escherichia coli* in their enteric contents [22]. The beneficial dietary effect of YLP on gut health has been linked to the presence of β-glucans in the yeast cell wall which protects the gastrointestinal tract against the colonization by dangerous pathogens, such us *Salmonella enterica* [20]. In support of this, previous studies have demonstrated that β-glucans and mannans, the two major components of the yeast cell wall, are bioactive components with potential benefits for the development of the intestinal immune system in animals [23]. They mitigate the release of pro-inflammatory cytokines and prevent the colonization of pathogens in intestinal mucosa.

According to a second explanation, the higher percentage of birds with no evidence of FPD recorded in the YLP groups, compared to the CON group, could also be due to the beneficial properties of the bioactive compounds of YLP for healthy skin, such as polyunsaturated fatty acids (PUFAs). YLP used in the present study consisted of approximately 39.7% PUFA, mainly linoleic acid (35.2%). It has been previously demonstrated that oils containing high levels of essential fatty acids improve skin hydration, regenerate the damaged epidermal lipid barrier and regulate skin metabolism [65]. Moreover, omega-3 and omega-6 FAs are significant components of the cell membrane, necessary for the function of the epidermal barrier; they display anti-inflammatory and anti-allergic effects by enhancing the repair processes and soothing irritation [65].

In the present study, the broilers of all experimental groups were evaluated with very good scores for hock burn, indicating no evidence of such a welfare issue. Regarding the feather cleanliness results, even though a higher percentage of birds with moderate feather soiling (Score 2) was recorded in the YLP5 group, compared to that observed in the CON

and YLP3 groups, this finding was not reflected in the FPD or hock burn outcomes. In a number of previous studies however, dirty feathers and FPD were highly correlated [66–68].

In current investigation, a significant antioxidant effect was detected in both thigh and breast meat of the broilers that were fed YLP diets, compared to the CON-fed birds, as indicated by the MDA results. This finding is very important because oxidation of the lipid components in muscle tissues is a major cause of quality deterioration and short shelf life after slaughter. The high concentration (42.5%) of oleic acid in YLP used in this study might have played a key role in this result. It has been previously demonstrated that oleic acid reduces oxidative stress and inflammatory response by activating the peroxisome proliferator-activated receptors in animals [69]. Complementarily, the antioxidant effect of YLP in broiler meat could also be attributed to the bioactive components of the yeast cell wall, such as β-glucans and mannans. Previous research evidence indicated that adding mannan oligosaccharides and β-glucans in poultry feed decreases the accumulation of the lipid oxidation end product (MDA) in the tissues of broiler chickens [70]. The beneficial dietary impact of YLP on the antioxidant status of turkey hens has been demonstrated in previous reports [16,18]. According to Merska et al. [16], supplementing turkeys' feed with 3% and 6% YLP yeast, increases the catalase activity (CAT) and decreases the plasma concentrations of lipid peroxidation products, such as hydroperoxide (LOOH) and MDA. Those results were confirmed in a later study by Czech et al. [18] who indicated that the role of YLP in the activation of antioxidant enzymes and the reduction of peroxidation products, when incorporated in turkeys' diet at 3% inclusion rate.

The supplementation of broilers' diet with 3% YLP yeast ameliorated the lipid profile of breast meat, as indicated by the increased concentration of MUFAs and PUFAs and the decreased levels of SFAs observed in this group. The reduced SFA concentration recorded in the breast meat of YLP3 birds could be primarily attributed to a reduction of palmitic acid, which was the most abundant among the SFAs recorded in the breast meat of all dietary treatments and, to a lesser degree, to a similar decrease of the stearic acid concentrations. The bioactive components of the yeast cell wall, such as β-glucans and mannans, might have played a key role to the reduction of the SFA levels in the breast meat of YLP3 birds, since they act as antioxidants and could suppress meat lipid oxidation. The increased level of PUFAs in the breast meat of the YLP3 group was mainly linked to an elevation of linoleic acid, the most abundant fatty acid among PUFAs found in the breast meat of all dietary treatments. Taking into consideration the high level of linoleic acid (35.2%) in the composition of the YLP used in the present study, it could be supported that the addition of YLP at a 3% incorporation rate in broilers' diet resulted in a concomitant increase of linoleic acid concentration in the breast meat of birds of this group. However, this beneficial dietary effect of YLP was not proportional to the incorporation rate of YLP in the birds' diet, as it was revealed by the breast lipid profile results of the YLP5 birds, which were similar with the CON group. It has been previously documented that the FA profile of chicken meat is affected by the birds' diet [71] and genetic factors [72]. Moreover, a FA diet composition, fat metabolism and fat deposition in edible tissues are often correlated in monogastric animals. Dietary supplementation with polyunsaturated FAs, such as linoleic, α-linolenic and arachidonic acids is often associated with increased levels of these acids in the muscle and adipose tissues, both through direct incorporation and modification of the unsaturated FA synthesis in these tissues [73,74]. However, the mechanisms involved are complicated and affect the lipogenic genes' expression [75,76].

The supplementation of broilers' diet with 3% YLP, positively affected the nutritional quality of their breast meat, as indicated by the improved PUFA/SFA ratio and the better health lipid indices recorded in the breast meat of YLP3 birds. The PUFA/SFA ratio is the most-commonly used index for estimating the effect of a certain food on cardiovascular health, considering that all PUFAs are capable to decrease low-density lipoprotein cholesterol and serum cholesterol, whereas all SFAs could contribute to the elevation of serum cholesterol [77]. Thus, higher values indicate a better (positive) effect given by a particular meat or meat product intake. The dietary ratios of PUFA to SFA, that are higher than

0.45, are regarded as safe for human consumption [78], and suitable to protect against the development of ischemic heart disease [79]. The optimal PUFA/SFA ratio from a nutritive point of view was achieved with the 3% incorporation rate of YLP, compared to the CON and YLP5 groups.

The most frequently used index for estimating the nutritional value of dietary foods is the PUFA/SFA ratio. However, it is regarded too general and inappropriate for evaluating the atherogenicity of foods [80] as specific SFAs and PUFAs have different metabolic effects [79]. Thus, AI was established [54] in order to estimate the atherogenic potential of the FAs in food. A lower value indicates better nutritional characteristics of the food [77]. Another important and commonly used index to further characterize the thrombogenic potential of FAs is TI [80]. This index points out the trend to form clots in blood vessels and indicates the contribution of different FAs, showing the relationship between the pro-thrombogenic FAs (C12:0, C14:0 and C16:0) and the anti-thrombogenic FAs (MUFAs and the n-3 and n-6 families) [54]. It has been documented that animal products with a low index of thrombogenicity reduce the threat of atrial fibrillation [81]. Summarizing, AI and TI can be employed for the estimation of the potential impact of the FA composition on cardiovascular health (CVH). An FA composition with a lower AI and TI, has a better nutritional quality, thus its consumption may decrease the risk of coronary heart disease (CHD), but the recommended values for the AI and TI have not been provided yet [80]. The results of this study indicated that the addition of 3% YLP in broilers' diet decreased significantly both the AI and TI, compared to the CON and YLP5 groups. At the same time, this incorporation rate of YLP yeast significantly reduced the h/H ratio of the produced breast meat. Santos-Silva et al. [82] supported that the higher the ratio between the hypocholesterolemic and the hypercholesterolemic fatty acids, the more the oil or fat is suitable to human nutrition. These findings indicate the health benefit potential related to the fat intake from breast meat produced from YLP3-fed broilers, since they presented a higher nutritional quality than the CON and YLP5 breast meat.

The FA profile analysis of the thigh meat revealed a significant elevation of n-3 PUFAs in the YLP-fed groups, compared with the CON group. This elevation is attributed to the higher concentration of α-linolenic acid observed in the thigh meat of the YLP groups, compared to the CON group, possibly related to the α-linolenic acid concentration of YLP used in the study. This finding is very important from the consumers' point of view. Intake of the recommended amounts of PUFAs, and particularly n-3 acids, is absolutely necessary for ensuring the correct functioning of the human body and essential for the prevention and mitigation of several diseases, such as cardiovascular diseases, skin diseases, autoimmune conditions and certain forms of cancer—breast, colon and prostate cancer [83–86]. Differences in the breast and thigh meat FA profile observed in the current study could be attributed to the different roles of FAs in these tissues or to their different phospholipid contents. The PUFAs are preferentially incorporated into phospholipids [87], and more phospholipids are in found in breast than in thigh muscles [88]. Since there is lack of research evidence regarding the effect of YLP in the lipid profile of breast and thigh meat, as well as on the relative health lipid indices when supplementing broilers' feed, the comparison of our results with those in literature is not possible. However, similar studies curried out on Atlantic salmon, have shown a beneficial effect of YLP on the FA composition in fish fillets [27].

In the present study, the response of the panelists for the breast and thigh meat of broilers fed diets supplemented with YLP yeast did not influence the sensory attributes of the broiler meat. However, according to the SET results, the thigh meat of YLP5 chicks scored the highest values for all sensory characteristics, leading to significantly higher overall acceptance scores, compared to the other dietary treatments. These findings are very important from the consumers' point of view, since they imply that YLP does not compromise the organoleptic characteristics of broiler meat; on the contrary, it increases the overall likeability of thigh meat. One of the main concerns when using foods rich in polyunsaturated lipids is the deterioration of the sensory quality of the poultry products. The results of several studies have demonstrated that the meat of animals containing more

PUFAs is more susceptible to oxidative processes [89], which has a negative impact on its organoleptic characteristics and shelf life [90]. In the present study however, despite the high concentrations of PUFAs in breast meat of the YLP3 group, as well as the higher concentration of n-3 PUFAs in the thigh meat of the YLP-fed broilers, compared to the CON group, the tenderness, juiciness, flavor, color and overall acceptance of YLP meat were not reduced. Instead, a positive effect on the overall acceptance of thigh meat from the YLP groups was detected with significant differences observed in the YLP5 group. This positive effect could be attributed to the antioxidant activity of oleic acid (present in high concentration 42.5%) in YLP used in this study, as well as to the bioactive components of the yeast cell wall, such as β-glucans and mannans, that have antioxidant properties protecting meat from lipid peroxidation. This is also supported by the lower values of MDA recorded in the breast and thigh meat of YLP birds, compared to the CON birds.

## 5. Conclusions

The present study, which is the first one carried out on broilers, shows that the supplementation of broilers' diet with dried YLP at two different levels (3% and 5%) positively affected FPD without any adverse effect on the overall growth performance of chicks, feather cleanliness and the qualitative behavior characteristics evaluated. These observations denote potential health and welfare benefits for YLP-fed chickens, while the producers can possibly benefit from a reduction of the production cost. These data also suggest that the broilers are able to adapt and efficiently consume diets containing YLP yeast at both concentrations tested (3% and 5%) at various growth stages. A meat quality analysis revealed a beneficial antioxidant dietary effect of YLP on both thigh and breast meat of YLP-fed broilers, as indicated by the reduced MDA values, compared to the CON group. This finding implies a potential beneficial effect of YLP to the quality and shelf life of meat after slaughter. Moreover, the supplementation of broilers' diet with 3% YLP yeast ameliorated the lipid profile of the breast meat, which also presented a superior nutrient quality, compared to the CON and YLP5 groups, as indicated by the better PUFA/SFA ratio, decreased values of AI, TI and the increased value of the h/H ratio. These observations demonstrate the health beneficial potential associated with the fat intake from breast meat produced from YLP3-fed broilers. Furthermore, an FA profile analysis of the thigh meat revealed a significant elevation in n-3 PUFAs in the YLP-fed groups, compared to the CON group. According to the SET results, tenderness, juiciness, flavor, color and the overall acceptance of YLP-fed broilers' meat were not reduced. Instead, a positive effect on the overall acceptance of thigh meat from YLP broilers was detected with significant differences observed in the YLP5 group. Finally, from the evaluated dietary levels of YLP, 3% seems to be more advantageous for the consumer, in terms of meat nutrition quality, so it is highly recommended.

**Supplementary Materials:** The following supporting information can be downloaded at: https://www.mdpi.com/article/10.3390/su15031924/s1, Figure S1: Schematic representation of the industrial-scale bioreactor pilot plant designed by NRRE Lab (CPERI/CERTH) used for the production of the yeast biomass. Table S1: Results of the quadratic regression for the relationships with broiler's overall acceptance (for all types of meat).

**Author Contributions:** Conceptualization, A.D., S.I.P. and E.N.S.; methodology, A.D., S.I.P., K.N.K., M.I. and A.Z.; software, A.D., M.-Z.K. and K.N.K.; validation, A.D., S.I.P. and E.N.S.; formal analysis, A.D. and K.N.K.; investigation, A.D., M.-Z.K. and M.I.; resources, S.I.P. and E.N.S.; data curation, A.D. and E.N.S.; writing—original draft preparation, A.D.; writing—review and editing, E.N.S., S.I.P. and K.N.K.; visualization, M.-Z.K., K.N.K., M.I. and A.Z.; supervision, E.N.S.; project administration, A.D., S.I.P., E.N.S. and A.Z.; funding acquisition, S.I.P. and E.N.S. All authors have read and agreed to the published version of the manuscript.

**Funding:** This research has been co-financed by the European Regional Development Fund of the European Union and Greek national funds through the Operational Program Competitiveness, Entrepreneurship and Innovation, under the call RESEARCH–CREATE–INNOVATE (project code: T1EDK-02871).

**Institutional Review Board Statement:** The study was conducted in accordance with the Declaration of Helsinki and approved by the Research Ethics Committee of the Hellenic Agricultural Organization DIMITRA (Reference Number 2340/57419/24-10-2022). The Research Ethics Committee of the Hellenic Agricultural Organization-DIMITRA has approved the experimental protocol and implemented animal care procedures of this study (Reference Number 2340/57419/24-10-2022).

**Data Availability Statement:** The data presented in this study are available upon request from the corresponding author. The data are not publicly available due to privacy.

**Acknowledgments:** We sincerely thank the DIMITRIADIS CHRISTOS S.A. poultry farm, who valuably contributed to this research by offering the experimental animals and facilities. Asimina Tsirigka (Chem. Eng. Dep., AUTh, Greece), Michail Sountourlis and Evangelos Tziaras (Fytoenergeia S.A., Serres, Greece) are greatly acknowledged for their technical contribution during the yeast biomass production process.

**Conflicts of Interest:** The authors declare no conflict of interest.

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
