# Peer review of "Growth Performance, Meat Quality, Welfare and Behavior Indicators of Broilers Fed Diets Supplemented with Yarrowia lipolytica Yeast"

_sustainability, doi:10.3390/su15031924_

Round 1

Reviewer 1 Report

-         The manuscript sustainability-2119847 entitled; “Growth performance, meat quality, welfare and behavior indicators of broilers fed diets supplemented with Yarrowia lipolytica yeast”. The authors investigate the evaluated the effects of Yarrowia lipolytica yeast on growth performance, meat quality, welfare, and behavioral indicators of broilers. In my opinion, the article has good data and a good presentation. Also, the experimental design is clear enough.

-         General comments

-         Please proofread the whole manuscript to avoid grammatical errors.

-         Please describe all abbreviations in their first mention.

-         In the discussion section, please be more specific, discuss your study with other similar studies and please state the superiorities of your study when compared to previous ones.

-         The conclusions section is too long, please rewrite it.

-         In the references section, use journal style. 

Reviewer 2 Report

Thank you very much for letting me review this article. I think, this is a wonderful article, nicely presented, hence must be processed further. I have made specific comments in different parts of the article, hence, please suggest authors to go through those.and address them accordingly.

Reviewer 3 Report

Thanks for your manuscript.

1. In my opinion, it is better to revise a word supplementation

* since the used levels reached 5%  ; it means it can

consider as feedstuff ingredients.

2. In Statistical analysis; it is better to add linear and quadratic and insert  inside tables, thus, 

 the authors can present the data clearly.

3.please when P≥0.05 delete a and b letters above means.

4. In table 6, please check the fat content in thigh meats; is it inside references range?  

5. In table 8, please use a letter a above the highest mean. 

6 . Ln 684 please add space between that the 
